# Challenges for BBVI with Normalizing Flows

**Akash Kumar Dhaka** [1]  **Alejandro Catalina** [1]  **Manushi Welandawe** [2]  **Michael Riis Andersen** [3]
**Jonathan Huggins** [2]  **Aki Vehtari** [1]

## Abstract

Current black-box variational inference (BBVI) methods require the user to make numerous design choices—such as the selection of variational objective and approximating family—yet there is little principled guidance on how to do so. We develop a conceptual framework and set of experimental tools to understand the effects of these choices, which we leverage to propose best practices for maximizing posterior approximation accuracy. Our approach is based on studying the pre-asymptotic tail behavior of the density ratios between the joint distribution and the variational approximation, then exploiting insights and tools from the importance sampling literature. We focus on normalizing flow models, though we are not limited to them.

## 1. Introduction

A great deal of progress has been made in black-box variational inference (BBVI) methods for Bayesian posterior approximation, but the interplay between the approximating family, divergence measure, gradient estimators and stochastic optimizer is non-trivial – and even more so for high-dimensional posteriors (Wang et al., 2018; Geffner & Domke, 2020; Agrawal et al., 2020). While the main focus in the machine learning literature has been on improving predictive accuracy, the choice of method components becomes even more critical when the goal is to obtain accurate summaries of the posterior itself.

We show that, while the choice of approximating family and divergence is often motivated by low-dimensional illustrations, the intuition from these examples do not necessarily

*Equal contribution [1]Department of Science, Aalto University Espoo, Finland [2]Department of Mathematics & Statistics Boston University, USA [3]DTU Compute, Technical University of Denmark. Correspondence to: Akash Kumar Dhaka <akash.dhaka@aalto.fi>, Aki Vehtari <aki.vehtari@aalto.fi>.

Third workshop on *Invertible Neural Networks, Normalizing Flows, and Explicit Likelihood Models* (ICML 2021). Copyright 2021 by the author(s).

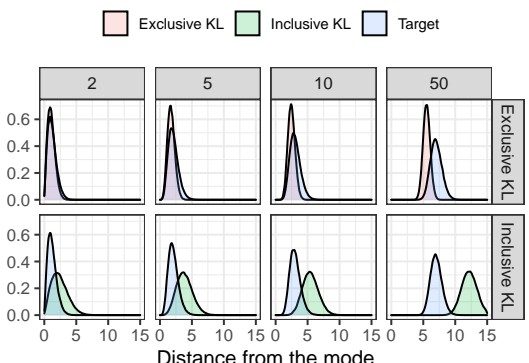

*Figure 1.* Illustration of a mean-field approximation with exclusive (mode-seeking) and inclusive (mass-covering) divergences. For correlated Gaussian targets in dimensions $D = 2, 5, 10, 50$, the marginal distributions of the distance from the mode for samples drawn from the approximation (red) and the target (blue). The intuition from the low-dimensional examples does not carry over to higher dimensions: although the importance ratios are still bounded, even for a lower correlation level, the overlap in typical sets of the target and the approximations gets worse both for exclusive and inclusive divergences.

carry over to higher-dimensional settings. By drawing a connection between importance sampling and the estimation of common divergences used in BBVI, we are able to develop a comprehensive framework for understanding the reliability of BBVI in terms of the *pre-asymptotic* behavior of the density ratio between the target and the approximate distribution. When this density ratio is heavy-tailed, even unbiased estimators exhibit a large bias with high probability, in addition to high variance. Such heavy tails occur when there is a mismatch between the typical sets of the approximating and target distributions. In higher dimensions, even over-dispersed distributions miss the typical set of the target (MacKay, 2003; Vehtari et al., 2019). Thus, as illustrated in Fig. 1, the benefits of heavy-tailed approximate families and divergences favoring mass-covering diminish as dimensionality of the target distribution increases. We develop a conceptual and experimental framework for predicting and empirically evaluating the reliability of BBVI based on the choice of variational objective, approximating family, and target distribution. Our framework also incorporates the Pareto $k$ diagnostic (Vehtari et al., 2019) as a simple and practical approach for estimating both the required minimal

sample size and obtaining empirical and conceptual insights into the pre-asymptotic convergence rates of estimators of common divergences and their gradients. Although normalizing flow models offer high capacity to cover posterior mass in high dimensions, they can be notoriously hard to optimize and our proposed framework is general and can be helpful to make design choices.

## 2. Preliminaries and Background

Let $p(\theta, Y)$ be a joint distribution of a probabilistic model, where $\theta \in \mathbb{R}^D$ is a vector of model parameters and $Y$ is the observed data. In Bayesian analysis, the posterior $p(\theta) := p(\theta \mid Y) = p(\theta, Y)/p(Y)$ (where $p(Y) := \int p(\theta, Y)d\theta$) is often the object of interest, but most posterior summaries of interest are not accessible because the normalizing integral, in general, is intractable. Variational inference approximates the exact posterior $p(\theta \mid Y)$ using a distribution $q \in \mathcal{Q}$ from a family of tractable distributions $\mathcal{Q}$. The best approximation is determined by minimizing a divergence $D(p \parallel q)$, which measures the discrepancy between $p$ and $q$:

$$q_{\lambda^*} = \arg\min_{q_\lambda \in \mathcal{Q}} D(p \parallel q), \qquad (1)$$

where $\lambda \in \mathbb{R}^K$ is a vector parameterizing the variational family $\mathcal{Q}$. Thus, the properties of the resulting approximation $q$ are determined by the choice of variational family $\mathcal{Q}$ as well as the choice of divergence $D$.

Let $w(\theta) := p(\theta, Y)/q(\theta)$ denote the density ratio between the joint and approximate distributions. For a function $\phi : \mathbb{R}^D \to \mathbb{R}$, the *self-normalized importance sampling estimator* for the posterior expectation $\mathbb{E}_{\theta \sim p}[\phi(\theta)]$ is given by

$$\hat{I}(\phi) := \sum_{s=1}^{S} \frac{w(\theta_s)}{\sum_{s'=1}^{S} w(\theta_{s'})} \phi(\theta_s),$$

where $\theta_1, \ldots, \theta_S \sim q$ are independent. Using importance sampling can allow for computation of more accurate posterior summaries and to go beyond the limitations of the variational family. Since importance sampling estimates can have very high variance, Pareto smoothed importance sampling (PSIS) can be used to substantially reduce the variance with small additional bias (Vehtari et al., 2019).

**Variational families.** Let $q_\lambda(\theta)$ be an approximating family parameterised by a $K$-dimensional vector $\lambda \in \mathbb{R}^K$ for $D$-dimensional inputs $\theta \in \mathbb{R}^D$. We focus on the most popular mean-field and normalizing flow families. Normalising flows provide more flexible families that can capture correlation and non-linear dependencies.

**$f$-divergences.** The most commonly used divergences are $f$-divergences. For a convex function $f$ satisfying $f(1) = 0$,

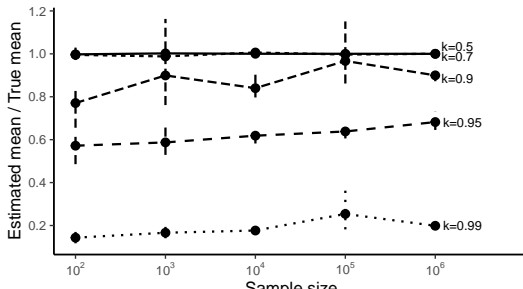

*Figure 2.* The ratio of estimated mean and true mean for different values of $k$ shape parameter of a generalized Pareto distribution and confidence intervals in a finite sample size simulation.

the $f$-divergence is given by

$$D_f(p \parallel q) := \mathbb{E}_{\theta \sim q}\left[ f\left( \frac{p(\theta \mid Y)}{q(\theta)} \right) \right].$$

The exclusive Kullback-Leibler (KL) divergence corresponds to $f(w) = -\log(w)$, the inclusive KL divergence corresponds to $f(w) = w\log(w)$, the $\chi^2$ divergence corresponds to $f(w) = (w-1)^2$, and the general $\alpha$-divergences correspond to $(w^\alpha - w)/\{\alpha(\alpha - 1)\}$. We also consider the *adaptive $f$-divergence* proposed by Wang et al. (2018).

**Loss estimation and stochastic optimization.** In all cases, minimizing the $f$-divergence is equivalent to minimizing the loss function $\mathcal{L}_f(p \parallel q) := \mathbb{E}_{\theta \sim q}[f(w(\theta))]$ (although, see Wan et al. (2020) for a different approach). Let $L(\lambda) := \mathcal{L}_f(p \parallel q_\lambda)$ denote the loss as a function of the variational parameters $\lambda$. The loss and its gradient $G(\lambda) := \nabla_\lambda L(\lambda)$ can both be approximated using the Monte Carlo estimates

$$\widehat{L}(\lambda) = \tfrac{1}{S}\sum_{s=1}^{S} f(w(\theta_s)) \quad \text{and} \quad \widehat{G}(\lambda) = \tfrac{1}{S}\sum_{s=1}^{S} g(\theta_s), \quad (2)$$

where $\theta_1, \ldots, \theta_S$ are independent draws from $q_\lambda$ and $g : \mathbb{R}^K \to \mathbb{R}^K$ is a gradient function that depends on $f$ and $w$. The gradient estimates can be used in a stochastic gradient optimization scheme such that

$$\lambda^{t+1} \leftarrow \lambda^t + \eta_t \widehat{G}(\lambda^t), \qquad (3)$$

where $\eta_t$ is the step size. In practice, more stable adaptive stochastic gradient optimisation methods such as RMSProp or Adam (Hinton & Tieleman, 2012) are often used.

## 3. Assessing the Reliability of BBVI

### 3.1. Conceptual framework

The most common variational divergences and their Monte Carlo gradient estimators can be expressed in terms of the density ratio $w(\theta)$. Reliable BBVI ultimately depends on the behavior of $w(\theta)$ since (1) accurate optimization requires low-variance and (nearly) unbiased gradient estimates $\widehat{G}(\lambda)$, and (2) determining convergence and validating the quality of variational approximations can require

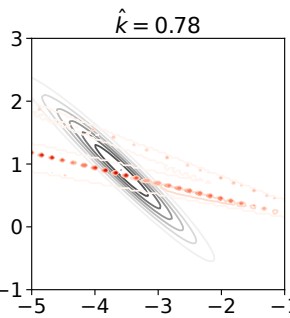

$\hat{k} = 0.78$

*Figure 3.* Wonky posterior approximation (in red) by NVP for robust regression heavy tailed posterior.

accurate estimates $\widehat{L}(\lambda)$ of variational divergences (Kucukelbir et al., 2015; Huggins et al., 2020). While *asymptotically* (in the number of iterations and Monte Carlo sample size $S$) there may be no issues with stochastic optimization or divergence estimation, in practice BBVI operates in the *pre-asymptotic* regime. **Therefore, the reliability of BBVI depends on the pre-asymptotic behavior of the $w(\theta)$, and how it interacts with the choice of variational objective and gradient estimator.** Before accounting for the effects of the objective and gradient estimator, first consider the behavior of the density ratio $w(\theta)$, which can also be interpreted as an importance sampling weight with $q_\lambda(\theta)$ as the proposal distribution (cf. Li & Turner, 2016; Wang et al., 2018; Bamler et al., 2017). Pickands (1975) proved, under commonly satisfied conditions, that for $u$ tending to infinity, the distribution of $w(\theta) \mid w(\theta) > u$ is well-approximated by the three-parameter generalized Pareto distribution $\mathsf{GPD}(u, \sigma, k)$, which for $k > 0$ has density $p(w \mid u, \sigma, k) = \sigma^{-1}\{1 + k(w - u)/\sigma\}^{-1-1/k}$ where $w$ is restricted to $(u, \infty)$. Since $w(\theta) > 0$, this implies its distribution is heavily skewed to the right with a power-law tail. Consider the idealized scenario of estimating the mean of $w(\theta) \sim \mathsf{GPD}(u, \sigma, k)$. We assume the mean is finite, which is equivalent to assuming $k < 1$ since $\lfloor 1/k \rfloor$ determines the number of finite moments. Because of the heavy right skew, most of the mass of $w(\theta)$ is below its mean. Therefore, even after averaging a large number of samples, most empirical estimates $\sum_{s=1}^{S} w(\theta_s)$ will be smaller than the true mean. Figure 5 illustrates this behavior for different values of $k$: even with 1 million samples, the empirical mean is far below the true mean when $k > 0.7$. The highly variable sizes of the confidence intervals based on 10,000 replications further highlight the instability of the estimator. **So, even though the empirical mean is an unbiased estimator, in the pre-asymptotic regime (before the generalized central limit theorem is applicable (Chen & Shao, 2004)), in practice the estimates are heavily biased downward with high probability.** If $w(\theta)$ is not a generalized Pareto distribution, we can instead treat $k$ as the *tail index* $k := \inf\{\ell > 0 : \mathbb{E}_{\theta \sim q}\{w(\theta)^{1/\ell}\} < \infty\}$, which encodes the same tail behavior as $\mathsf{GPD}(u, \sigma, k)$. Crucially,

we should expect $k$ to be much larger than 0 when there is a significant mismatch between the target distribution and the variational family. **Since selecting a variational family that can match the typical set tends to be more difficult in higher dimensions, we should expect $k$ to be larger for higher-dimensional posteriors.**

We can generalize our observations about pre-asymptotic estimation bias to the estimators $\widehat{L}(\lambda)$ and $\widehat{G}(\lambda)$. For the loss estimator, we replace $w(\theta_s)$ with $f(w(\theta_s))$, where $f(w)$ is polynomial in $w$ and $\log w$ for the class of losses we consider. If the dominant term of $f(w)$ is of order $w^\alpha$, the tail behavior will be similar to a generalized Pareto with $k_\alpha = \alpha k$. Thus, $\widehat{L}(\lambda)$ will have larger pre-asymptotic bias as $\alpha$ increases. For example, estimation of the mass-covering inclusive KL (where $\alpha = 1$) – and, more generally, mass-covering $\alpha$-divergences with $\alpha > 0$ – will suffer from a large pre-asymptotic bias. On the other hand, for the mode-seeking exclusive KL, $f(w) = \log(w)$, so we can expect all moments to be finite and therefore a much smaller pre-asymptotic bias.

Similar considerations apply to the gradient estimator. However, when using self-normalized weights for $\alpha$-divergences, we can expect a large pre-asymptotic bias whenever $w(\theta)$ has such bias since self-normalization involves estimating the mean of $w(\theta)$. This bias will affect the accuracy of the solution found. The quality of the solutions found can only partially be improved by using a smaller step size, since smaller step sizes will only reduce the effects of a large estimator variance, but not the effects from a large bias.

### 3.2. Experimental framework

In the light of potentially large non-asymptotic bias arising from the heavy right tail of $w(\theta)$, it is important to verify the pre-asymptotic behavior of the Monte Carlo estimators used in variational inference. We follow the approach developed by Vehtari et al. (2019) for importance sampling and compute an empirical estimate $\hat{k}$ of the tail index $k$ by fitting a generalized Pareto distribution to the observed tail draws. Vehtari et al. (2019) show that the minimal sample size to have a small error with high probability scales as $S = \mathcal{O}(\exp\{k/(1-k)^2\})$. Two key propositions here are: **(P1) Mode-seeking divergences are more stable and reliable than mass-covering ones.** Fig. 8 in Appendix shows that gradient bias and variance increases with dimension for inclusive KL and $\chi^2$ but not exclusive KL.

**(P2) Degree of polynomial dependence on $w$ determines sensitivity to approximation–target mismatch.** Fig. 8 also shows that gradient estimates (divergence estimates in Appendix A.1) of divergences that involve higher polynomials of $w$ (e.g. $\chi^2$) become more and more unstable as dimensions increase.

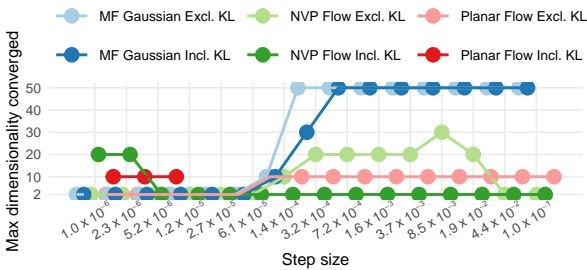

Figure 4. Maximum dimensionality converged per step size for the robust regression model.

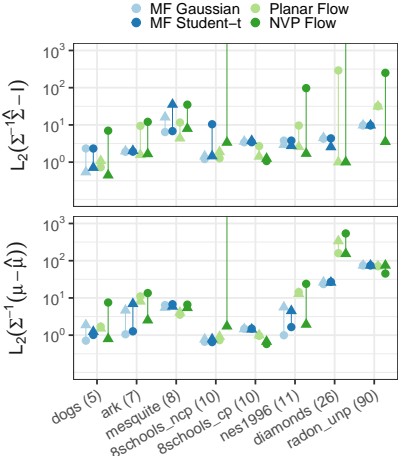

Figure 6. Relative error of mean and covariance estimates for BBVI using exclusive KL (circles) and after PSIS correction (triangles).

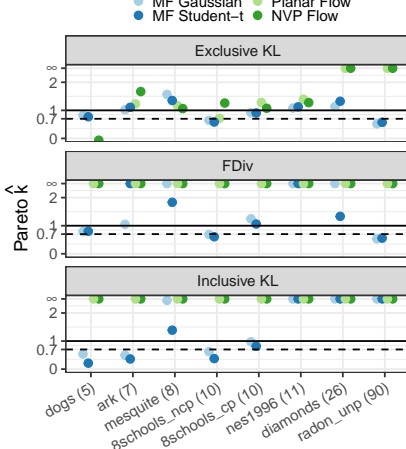

Figure 5. Results for `posteriordb` experiments. Dimensionality of each dataset is given in parentheses. Pareto $\hat{k}$ values for BBVI approximations.

## 4. Experiments

In this section, we describe a series of experiments to show how our framework works. For all posteriors, we fit mean-field Gaussian and Student-$t$ families, a planar flow (Rezende & Mohamed, 2015) with 6 layers and a non-volume preserving (NVP) flow (Dinh et al., 2017) with 6 stacked neural networks with 2 hidden layers of 10 neurons each for both the translation and scaling operations. We use Stan (Stan Development Team, 2020) for model construction.

**Mode-seeking divergences are easier to optimize.** In particular, the $\hat{k}$ values when using normalizing flows, which are more challenging to optimize, is low for $D < 20$ when using exclusive KL, but infinite when using either the inclusive KL or $f$-divergence. We can see that exclusive KL provides also more accurate and reliable posterior approximations than the inclusive KL and adaptive $f$-divergence, particularly for the normalizing flows. This is despite the fact that we used 20 times as many MC samples to estimate the gradients for the inclusive KL and the $f$-divergence compared to the exclusive KL. To further illustrate the relative difficulty of optimizing the inclusive KL divergence,

Fig. 4 shows the largest dimension for which the stochastic optimization converged as a function of the step-size. For most step-sizes, the combination of normalizing flows and the inclusive KL divergence only converged for $D = 2$, whereas convergence is possible in higher dimensions for simpler variational families.

### 4.1. Realistic models and datasets

We compare variational approximations for models and datasets from `posteriordb` (github.com/stan-dev/posteriordb) in terms of accuracy of the estimated moments and predictive likelihood. We use $T_{\max} = 15,000$ and robust optimisation algorithm given in Dhaka et al. (2020) for stochastic optimisation, for cases where the divergence objectives are analytical, we used BFGS optimisation.

**Exclusive KL remains the most reliable for realistic posteriors.** The results are summarized in Figs. 5 and 6: exclusive KL is superior for higher-dimensional posteriors (e.g., $D > 10$) or when combined with normalizing flows, while inclusive KL is better for the large values for $\hat{k}$ indicate that fitting approximations based on normalizing flows remains a challenge in high dimensions. The performance for the adaptive $f$-divergence is comparable to the inclusive KL divergence. **Normalizing flows can be effective but are challenging to optimize.** Fig. 6 also shows that normalizing flows can be quite effective when used with exclusive KL to ensure stable optimization However, as can be seen in Fig. 5 and Fig. 3, when using out-of-the-box optimization with no problem-specific tuning (as we have done for a fair comparison), the normalizing flows approximations can have pathological features – even in low dimensions.

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

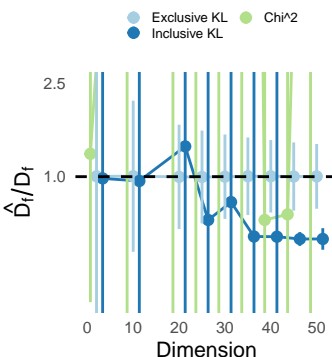

*Figure 7.* Results for the ratio of the $f$-divergence estimate to the true value of a correlated Gaussian targets of dimension $D = 1, \ldots, 50$ using either the exclusive or inclusive KL divergence as the variational objective.

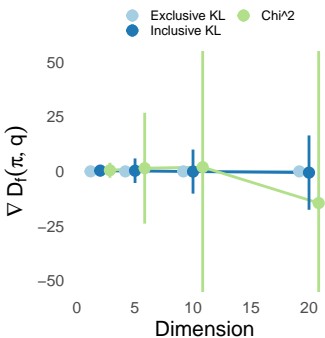

*Figure 8.* The bias and variance of the gradients of the different divergence objectives for one variational parameter at the end of optimisation for correlated Gaussian targets of dimension $D = 1 \cdots 20$

*Table 1.* Datasets from PosteriorDB.

| Name | Dimensions |
|---|---|
| Dogs | 5 |
| Ark | 7 |
| Mesquite | 8 |
| Eight schools non centered | 10 |
| Eight schools centered | 10 |
| NES1996 | 11 |
| Diamonds | 26 |
| Radon unpooled | 90 |

mising $\chi^2$, $1/2$-divergence and tail adaptive $f$-divergence follow similar trends as those resulting from optimising exclusive and inclusive KL. Approximations obtained by optimising $\chi^2$ and $1/2$-divergence are more unstable and end up diverging in similar ways as inclusive KL even for moderately low dimensional problems. We use a warm start procedure for $\chi^2$, $1/2$-divergence and inclusive KL, starting at the solution of exclusive KL for a given problem. On the other hand, optimising tail adaptive $f$-divergence seems to be more robust and behave similarly to exclusive KL even in higher dimensions.

## A. Appendix

### A.1. Divergence estimates

We add an additional visualization that demonstrates the unstable behaviour of the $f$-divergence estimates as dimensions increase, particularly for divergences that involve higher polynomials of $w$.

## B. PosteriorDB datasets

In Table 1 we show the dimensionality of the datasets we use for our real experiments.

## C. Additional results for the pre-asymptotic reliability case study

In Fig. 9 and Fig. 10 we show additional results for the pre-asymptotic reliability case study for different objectives and mean field Gaussian approximation. The results from opti-

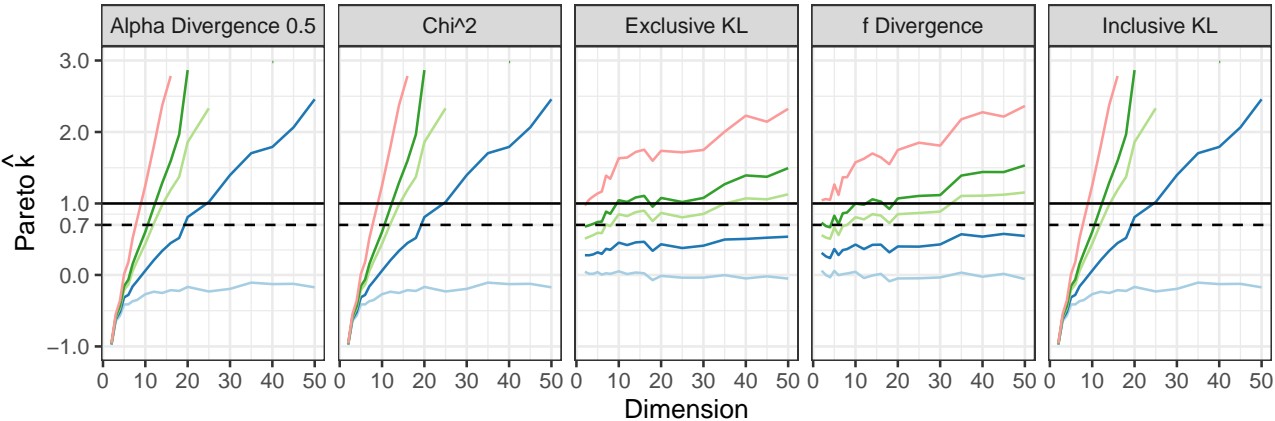

*Figure 9.* Pareto $\hat{k}$ estimated for different objectives and divergences estimation for a $0.5$ correlated Gaussian target and mean field Gaussian approximation and increasing dimensionality. Here we compute the $\hat{k}$ for all the f(w) after optimizing a particular variational objective.

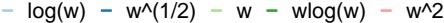

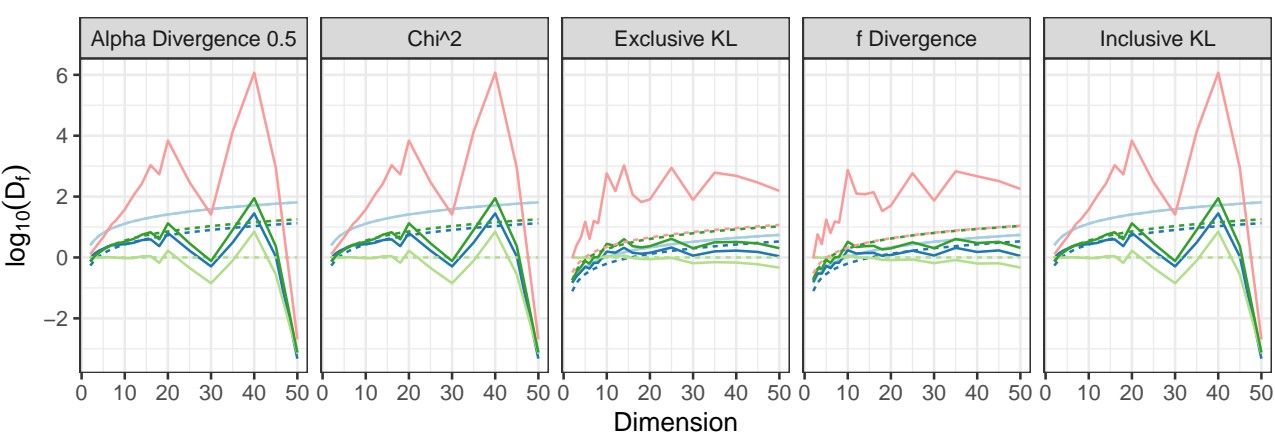

*Figure 10.* Divergences estimates for different objectives for a $0.5$ correlated Gaussian target and mean field Gaussian approximation and increasing dimensionality.