# OpenReview forum: "Challenges for BBVI with Normalizing Flows"
_ICML.cc/2021/Workshop/INNF — INNF+ 2021 poster_

### Official Review · Reviewer_LetF · 2021-06-08

**Rating:** Borderline Accept
**Confidence:** 3

**Summary:**

The paper analyses different types of divergencies from an optimization perspective. It draws conclusions based on the proposed conceptual framework for theoretical analysis and verifies them experimentally.

**Justification For Rating:**

The considered problem is important, and the proposed framework seems useful, though, I cannot comment on its novelty. There are the following comments which make my rating the borderline accept:
* Section 4 is underdeveloped and hard to follow.
** It seems there is a missing reference to a figure in the first half of the "Mode-seeking divergences are easier to optimize" part (line 217 says we can see ).
** There is no description of datasets used, only the reference to a repository.
** What is the PSIS correction mention in Fig. 7?
** what do dashed lines represent in Fig. 7?
* The statement in lines 126-128 (right) about the tail behavior might need justification, doesn't seem to be obvious without a reference or proof.
* The conclusion that would summarize the findings describing "best practices for maximizing posterior approximation" mentioned in the abstract would be helpful and is missing now.

Minor typos:
* line 143 left: it seems there should be the probability sign
* line 156 left: should there be the reference to Fig. 3?


**[Optional] Respond To Feedback Request By The Authors:**

It would be interesting to see such an analysis for Bayesian Neural Netwroks.

---

### Official Review · Reviewer_vZo6 · 2021-06-11

**Rating:** Borderline Accept
**Confidence:** 2

**Summary:**

This paper proposes an analysis of Black Box Variational Inference which mainly considers the pre-asymptotic regime of the MC estimators involved in optimizing the loss via stochastic optimization. The framework allows to draw some conclusions regarding the choice of the divergence to be minimized as well as how to tune optimization hyper-parameters.

**Justification For Rating:**

While I found there were many interesting and thought-provoking ideas in this work, including interesting propositions for the experiments  and the fact that the analysis accounts for pre-asymptotic behavior, I think the writing is hard to follow in some places and the paper would benefit from a better organization.

Moreover, the experiments section is frustratingly short and the consequences brought by the analysis in the proposed framework should be explored more (even though I understand that this is a short workshop submission).

---

### Official Review · Reviewer_c5Lt · 2021-06-12

**Rating:** Borderline Reject
**Confidence:** 2

**Summary:**

The quality of posterior approximation accuracy in Black-Box Variational Inference (BBVI) depends on the choice of the approximating family of distributions (e.g. normalizing flows) and variational objective.
This work proposes a framework incorporating the $k$ Pareto diagnostic to study this problem, seeking to provide guidance for these design choices.

**Justification For Rating:**

The considered problem, that is—assessing and evaluating the reliability of BBVI depending on specific modeling choices, is of potential interest.
In its current state, however, it is unclear what the key contribution of the submission is.
How exactly practitioners are supposed to benefit from the proposed framework is not explained very clearly. If the framework is supposed to provide a basis for empirical evaluation of BBVI with normalising flows and other methods, and using different variational families, stronger experiments illustrating this would be required.
The discussion of the experiments, particularly in section 4.1 and on normalising flows, demands more elaboration. In general, clarity could be improved throughout.

Typos:

Figure 3 has the same caption as Figure 2.

Typo in Figure 6 Caption — “parentheses.Pareto”, unspaced

Section 4.1: “optimization However”, dot missing

**[Optional] Respond To Feedback Request By The Authors:**

Adding a "Conclusions" section would be useful. Improving clarity, adding a stronger experimental validation and possibly an appendix expanding on the details of the framework presented in section 3.1 would, in my view, strengthen the submission.

---

### Decision · Program_Chairs · 2021-06-15

**Decision:**

Accept (poster)

**Comment:**

The paper topic is within the scope of the workshop. All reviewers agree that the topic of assessing the effects of the different modeling choices in BBVI is important. The reviewers are of the opinion that the conclusion of the analysis of the paper could be improved, so that it is clear what exactly are the best practices for maximizing posterior approximation, according to this paper. Furthermore, the experiment section could benefit from rewriting and more details. This paper is accepted to the workshop, but we urge the authors to take into account the feedback from the reviewers to improve their paper.